# Dehallucinating Parallel Context Extension for Retrieval-Augmented Generation

## Abstract

Large language models (LLMs) are susceptible to generating hallucinated information, despite the integration of retrieval-augmented generation (RAG). Parallel context extension (PCE) is a line of research attempting to effectively integrating parallel (unordered) contexts, while it still suffers from in-context hallucinations when adapted to RAG scenarios. In this paper, we propose **DePaC** (**De**hallucinating **Pa**rallel **C**ontext Extension), which alleviates the in-context hallucination problem with **context-aware negative training** and **information-calibrated aggregation**. DePaC is designed to alleviate two types of in-context hallucination: **fact fabrication** (i.e., LLMs present claims that are not supported by the contexts) and **fact omission** (i.e., LLMs fail to present claims that can be supported by the contexts). Specifically, (1) for fact fabrication, we apply the context-aware negative training that fine-tunes the LLMs with negative supervisions, thus explicitly guiding the LLMs to refuse to answer when contexts are not related to questions; (2) for fact omission, we propose the information-calibrated aggregation which prioritizes context windows with higher information increment from their contexts. The experimental results on nine RAG tasks demonstrate that DePaC significantly alleviates the two types of in-context hallucination and consistently achieves better performances on these tasks.

## 1 Introduction

Retrieval-augmented generation (RAG) (Lewis et al., 2020; Gao et al., 2023) is nowadays a prevalent paradigm for incorporating large language models (LLMs) (OpenAI, 2023; Touvron et al., 2023; Jiang et al., 2023a) with outside knowledge. RAG employs a *retriever* to fetch documents that are semantically closest to the question, and incorporates them into LLM's prompt. Parallel Context Extension (PCE) (Hao et al., 2022; Ratner et al., 2023; Su et al., 2024) is a line of research attempting to effectively integrating parallel contexts through an aggregation function. PCE is highly compatible with RAG scenarios, as the candidate retrieved documents of RAG are independ of each other.

However, existing PCE approaches still face two types of in-context hallucination issues (Ji et al., 2023; Rawte et al., 2023; Yang et al., 2023): **fact fabrication** and **fact omission.** (1) **fact fabrication** occurs when the model presents fabricated claims that are inconsistent with the contextual facts. As shown in Figure 2a, LLM confidently produces a fabricated answer for the window with $Doc_2$, caused PCE to fabricate the wrong answer. (2) **fact omission** refers to windows lacking useful information may disproportionately affect the aggregation function, leading it to omit critical information present in other windows. This will make LLMs fail to present claims that can be supported by the contexts. As shown in Figure 2b, $Doc_3$ does not contain required information, makes LLM confidently generate *"Unknown"* for the window with $Doc_3$, further leading to the wrong final answer.

In this paper, we propose DePaC to alleviate the hallucination issue of parallel context extension on RAG scenario. DePaC contains two parts: **NegTrain** (Context-aware **Neg**ative **Train**ing) to address fact fabrication issue and **ICA** (**I**nformation-**C**alibrated **A**ggregation) to address fact omission issue. (1) **NegTrain** guides the LLMs to refuse to answer when contexts are not related to the question. NegTrain consists of two parts of training data: one part comprises useful documents and questions as input, with corresponding answers as output. While the other part treats irrelevant documents and questions as input, with a rejection token as output. (2) **ICA** prioritizes context windows with higher information increment from their contexts. Specifically, we utilize Kullback-

Leibler divergence (Kullback & Leibler, 1951) to measure the information increment of with-document compared to non-document. This approach enhances DePaC's capability to identify useful information within parallel windows. Moreover, DePaC has lower computational complexity than vanilla inference approach. The inference time of DePaC increases linearly with the number of documents, while inference time of vanilla approach increases quadratically.

We conduct experiments on various RAG tasks, demonstrate that DePaC significantly alleviates the two types of hallucinations and consistently achieves promising performances. Then we analyze the proportion of hallucination produced by different approaches, demonstrating that DePaC can effectively mitigate the two types of hallucinations (Figure 1). We also conduct ablation study to identify that information-calibrated aggregation and context-aware negative training are both essential for DePaC performance.

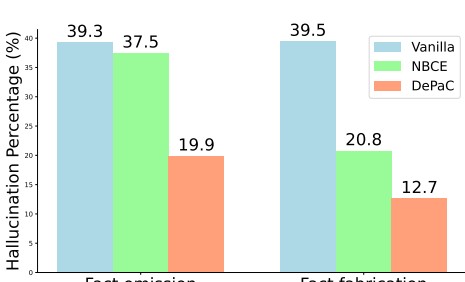

Figure 1: DePaC significantly reduces the occurrence of hallucinations in responses within RAG scenarios.

The main contents of this paper are organized as follows. Section 2 introduces the formalization of PCE and two existing aggregation methods for PCE. Section 3 introduces the methodology and implementation details of DePaC. Section 4 introduces the complexity analysis of DePaC. Section 5 introduces our experimental results on information seeking and DocQA. Section 6 discusses the related work. Finally, section 7 provides a conclusion regarding our work.

## 2 BACKGROUND: PARALLEL CONTEXT EXTENSION (PCE)

The core idea of PCE involves aggregating information from multiple context windows into a unified representation space. Such a representation aggregation can be formalized on either the probability distributions of output tokens (Su et al., 2024), or the internal hidden states in attention layers (Hao et al., 2022; Ratner et al., 2023). Su et al. (2024) claimed the above two formalizations have similar practical performances. In this work, we adopt the formalization in (Su et al., 2024) that takes the aggregation of output distributions.

Given an question $\mathcal{Q}$, a set of retrieved documents $\mathcal{D} = \{d_1, d_2, ..., d_n\}$, and a language model with parameters $\theta$, PCE first computes the output distribution of each context window,

$$\mathbf{p_{i,j}} = p_\theta(\,\cdot\,|\,d_j \oplus \mathcal{Q} \oplus \mathcal{A}_{1:i-1}), \tag{1}$$

where $\mathbf{p_{i,j}}$ is the probability distribution of the $i$-th token for output $\mathcal{A}$ based on the $d_j$ document, and $\oplus$ represents the concatenation of sequences. Subsequently, these individual distributions are aggregated into a single distribution,

$$\mathbf{p_i} = \mathrm{AGG}(\mathbf{p_{i,1}},\ \mathbf{p_{i,2}},\ ...,\ \mathbf{p_{i,n}}), \tag{2}$$

where $\mathrm{AGG}(\cdot)$ represents the aggregation method. Finally, the output token $\mathcal{A}_i$ will be sampled based on the aggregated distribution $\mathbf{p_i}$,

$$\mathcal{A}_i \sim \hat{\mathbf{p_i}}, \quad \hat{\mathbf{p_i}} = \mathbf{p_i} - \alpha \cdot \mathbf{p_{i,c}}, \tag{3}$$

$$\mathbf{p_{i,c}} = p_\theta(\,\cdot\,|\,\mathcal{Q} \oplus \mathcal{A}_{1:i-1}), \tag{4}$$

where the $\hat{\mathbf{p_i}}$ is the calibrated distribution to facilitate generation. We set $\alpha = 0.2$ following Su et al. (2024).

The effectiveness of the PCE paradigm is significantly influenced by the design of the aggregation method $\mathrm{AGG}(\cdot)$. Here, we discuss two aggregation methods used in existing studies.

**Average Aggregation** (Hao et al., 2022; Ratner et al., 2023). The aggregated distribution is computed as the average of $n$ individual distributions,

$$\mathbf{p_i} = \frac{1}{n} \sum_{j=1}^{n} \mathbf{p_{i,j}}. \tag{5}$$

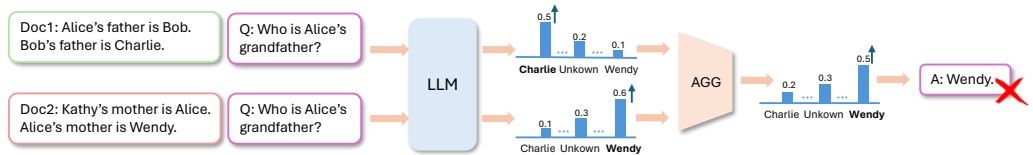

(a) Fact fabrication example. $Doc_2$ is useless to answer the question. The higher confidence in *"Wendy"* on $Doc_2$ caused PCE to fabricate the answer "Alice's grandfather is Wendy."

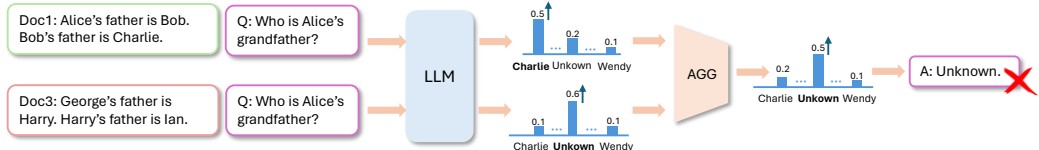

(b) Fact omission example. $Doc_3$ is useless to answer the question. The higher confidence in "unknown" on $Doc_3$ caused PCE to omit the fact on $Doc_1$, resulting an incorrect final answer after aggregation.

Figure 2: Existing PCE approaches face two types of in-context hallucination issues when applied to RAG: (1) Fact fabrication. LLM generates fabricated answers that are inconsistent with the contextual facts. (2) Fact omission. The absence of required information in certain windows disproportionately influence the aggregation function, leading to disregard critical information in other windows.

In practice, the size of the retrieved document set $\mathcal{D}$ can be large, potentially containing only a few relevant documents. Average aggregation treats each context window with equal importance, makes it unable to seek critical information when applied to RAG.

**Lowest-Uncertainty Aggregation** (Su et al., 2024). This method selects the individual distribution with the lowest uncertainty as the aggregation result,

$$\mathbf{p_i} = \arg\min_{\mathbf{p_{i,j}}} H(\mathbf{p_{i,j}}), \tag{6}$$

$$H(\mathbf{p_{i,j}}) = -\mathbf{p_{i,j}}(\log \mathbf{p_{i,j}})^T. \tag{7}$$

Lowest-uncertainty aggregation addresses the limitations of average aggregation by filtering out high-uncertainty windows. However, it remains a sub-optimal solution as it still suffers from the two types of hallucinations illustrated in Figure 2.

## 3 Dehallucinating Parallel Context Extension (DePaC)

As shown in Figure 3, we propose two methods to alleviate the fact fabrication and fact omission hallucinations of PCE for RAG scenarios. First, we introduce **Context-aware Negative Training** to enable the model to refuse to answer questions when the relevant information is missing in the context, thereby mitigating fact fabrication. Then, we propose **Information-Calibrated Aggregation** to measure the information increment given by the document, preventing the model from fact omission.

**Context-aware Negative Training (NegTrain).** We introduce context-aware negative training to alleviate fact fabrication, which explicitly train the backbone model to determine whether a question is answerable based on the provided document. If not, we hope the model to refuse to answer the question rather than generating hallucinations.

Given an RAG example with a question $\mathcal{Q}$, a ground-truth answer $\mathcal{A}$, and a retrieved document $d_j$, we fine-tune the backbone model $\theta$ according to the following loss function,

$$\text{Loss}(\mathcal{Q}, \mathcal{A}_{1:m}, d_j) = \begin{cases} \text{CE}[p_\theta(\,\cdot\,|\,d_j \oplus \mathcal{Q}),\ \mathcal{A}_{1:m}], & \text{related}(\mathcal{Q}, d_j), \\ \text{CE}[p_\theta(\,\cdot\,|\,d_j \oplus \mathcal{Q} \oplus \mathcal{A}_{1:i}),\ t_d], & \text{else}, \end{cases} \tag{8}$$

where $\text{CE}[\cdot]$ represents the cross-entropy loss, $t_d$ is a pre-defined **rejection token**, $m$ refers to the sequence length of the ground-truth answer, $\mathcal{A}_{1:m}$ refers to the complete ground-truth answer with all tokens, $\mathcal{A}_{1:i}$ refers to the partial ground-truth answer the first tokens. As shown in Figure 3(1), to

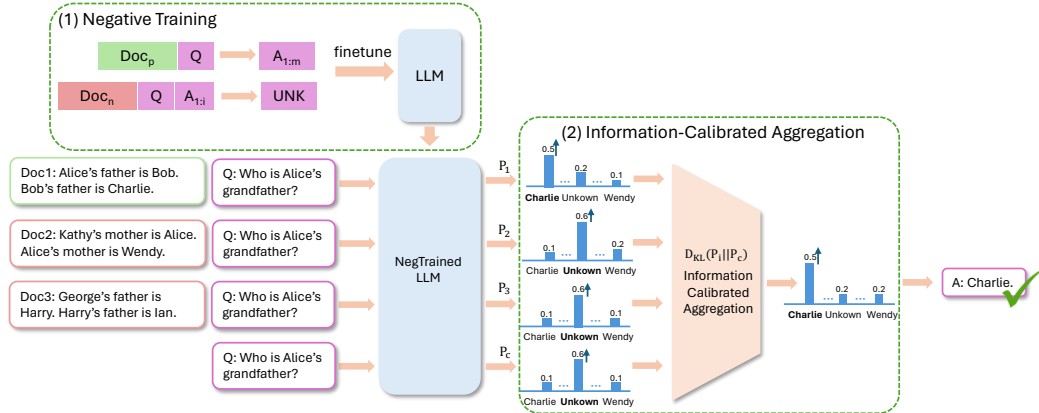

Figure 3: DePaC consists of two key components: (1) a context-aware negative training technique to alleviate fact fabrication, and (2) an information-calibrated aggregation method to alleviate fact omission.

prevent DePaC from generating rejection token only at the beginning of the answer, we also include the positive answer clauses as input. After context-aware negative training, we use $t_d$ to explicitly judge the usefulness of each context window. We set $t_d$ as the UNK token to minimize interference with normal tokens during training.

**Information-Calibrated Aggregation (ICA).** As discussed in Section 2, merely measuring the uncertainty of the final output distribution can be heavily influenced by fact omission hallucination. We propose to measure the changes of uncertainty from the non-document output distribution to the with-document output distribution, reflecting the information increment provided by the retrieved document.

Specifically, we apply the Kullback-Leibler (KL) divergence to measure the information increment,

$$\Delta(\mathbf{p_{i,j}}, \mathbf{p_{i,c}}) = D_{KL}(\mathbf{p_{i,j}} \,||\, \mathbf{p_{i,c}}), \tag{9}$$

$$\mathbf{p_{i,c}} = p_\theta(\,\cdot\,|\,\mathcal{Q} \oplus \mathcal{A}_{1:i-1}), \tag{10}$$

where $\mathbf{p_{i,c}}$ is the non-document output distribution.

Finally, we integrate the above two methods as two penalty terms to inject into Equation 6,

$$\mathbf{p_i} = \arg\min_{\mathbf{p_{i,j}}}[C(\mathbf{p_{i,j}}, \mathbf{p_{i,c}}) - \gamma \cdot \mathbb{I}(\arg\max_k \mathbf{p_{i,j}}^k = t_d)], \tag{11}$$

$$C(\mathbf{p_{i,j}}, \mathbf{p_{i,c}}) = H(\mathbf{p_{i,j}}) - \beta \cdot \Delta(\mathbf{p_{i,j}}, \mathbf{p_{i,c}}), \tag{12}$$

where $\mathbb{I}[\cdot]$ represents the indicator function, $\mathbf{p_{i,j}}^k$ is the output probability on $k$-th token in the vocabulary, and $\beta > 0$ and $\gamma < 0$ are hyper-parameters. Equation 11 and 12 mean that the selected context window should have low uncertainty and high information increment, and should not be aligned to the rejection token. Finally, the output token $\mathcal{A}_i$ will be sampled based on the aggregated distribution $\mathbf{p_i}$. For ease of implementation, we provide a simplified form of DePaC in Appendix B.

**Implementation Details** Following previous work (An et al., 2024), we use the C4 (Raffel et al., 2020) corpus to construct our context-aware negative training dataset. For a segment of text from C4, we first split it into text fragments with a maximum length of 4k tokens. We first sample a fragment serves as oracle document, and use GPT-4-Turbo to generate questions and answers based on the oracle document as positive training data. Then we sample unrelated fragment serves as distractor document to construct context-aware negative training data based on the positive ones. To prevent the model from overfitting on $t_d$, we control $t_d$ occurrence to match the average frequency of the 2,000 most frequent tokens in NegTrain. Finally, we construct 19K samples for context-aware negative training. We fine-tune three open-source models (introduce in Section 5.3) using 8x80G A100 GPUs, set the global batch size as 128 and trained for two epochs. We use Flash Attention-2 (Dao, 2023) to enhance the training speed. The entire training process takes about 4 hours.

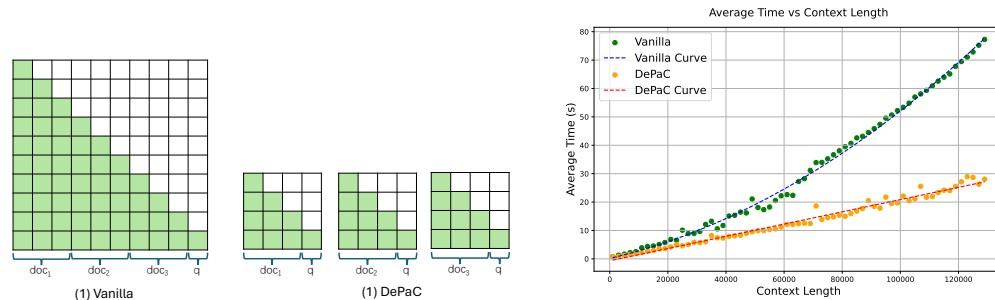

Figure 4: Attention pattern and execution time comparison between DePaC and vanilla inference. The execution time of DePaC increases linearly with context length, while vanilla's complexity grows quadratically.

## 4    COMPLEXITY ANALYSIS

Considering that RAG scenarios have high expectations for execution efficiency and previous PCE-style work lacked analysis of the execution efficiency, we present the inference complexity of DePaC compared with vanilla inference approach. Figure 4 shows the attention pattern and execution time comparison between DePaC and vanilla inference. As the length of the question is much smaller than the length of the document, the complexity of processing the question is ignored. Given a LLM with $m$ layers, we assume that the context consists of $k$ documents, each with $n$ tokens.

**Vanilla complexity.**    Vanilla inference directly concatenates the $k$ documents as the input to LLM, with a sequence length of $kn$. The attention of each layer is calculated by $\text{Attention}(Q, K, V) = \text{softmax}\left(QK^T\right)V$, where $Q, K, V \in \mathbb{R}^{(kn) \times d}$ is the query, key and value matrix. The complexity of $QK^T$ is $\mathcal{O}((kn)^2 \cdot d)$. So the complexity of $Attention(Q, K, V)$ for $m$ layers is $\mathcal{O}(k^2 \cdot n^2 \cdot d \cdot m)$.

**DePaC complexity.**    In DePaC, $k$ documents are inputted to LLM in parallel, the sequence length for each input is $n$. This is akin to $k$ times $Attention(Q, K, V)$ computations, but with smaller $Q, K, V \in \mathbb{R}^{n \times d}$, so the complexity of $Attention(Q, K, V)$ for $m$ layers is $\mathcal{O}(k \cdot n^2 \cdot d \cdot m)$.

The complexity of Vanilla increases quadratically with $k$, while DePaC's complexity grows linearly. Figure 4 shows the average execution time of DePaC and vanilla inference approach with different context length, DePaC has faster inference speed than vanilla approach. Moreover, DePaC can place all documents in a single batch for parallel processing, further enhancing DePaC's inference speed.

## 5    EXPERIMENTS

We conduct experiments on various tasks to assess DePaC's performance on RAG and alleviate the two types of in-context hallucination.

### 5.1    TASKS

We conduct evaluations on nine RAG tasks, including six information seeking tasks and three document-based question-answering tasks.

The **information seeking** tasks serve to explicitly probe the information awareness of DePaC. Each test case in these tasks contains an information query question and a large amount of contexts. Based on the given question, the model is required to seek for some textual pieces within the contexts. The information seeking tasks include: Function name retrieve **(FuncNR)** (An et al., 2024), Entity label retrieve **(EntLR)** (An et al., 2024), Multi-values Needle-in-a-Haystack **(MVIH)** (Hsieh et al., 2024), TensorHub APIBench**(Tens)** (Patil et al., 2023), TorchHub APIBench**(Torc)** (Patil et al., 2023), and Huggingface APIBench**(Hugg)** (Patil et al., 2023). Appendix  C shows the detailed description of information seeking tasks.

Table 1: Comparison of DePaC with baselines across three models and nine tasks.

| Model | Method | FuncNR | EntLR | MVIH | Tens | Torc | Hugg | Qasper | MulQA | NarQA | Avg |
|-------|--------|--------|-------|------|------|------|------|--------|-------|-------|-----|
| Mistral-7B | Vanilla (Jiang et al., 2023a) | 25.4 | 44.1 | 21.9 | 37.1 | 14.5 | 1.4 | 15.0 | 39.7 | 10.2 | 23.3 |
| | AVP (Hao et al., 2022) | 2.3 | 0.3 | 0.3 | 38.8 | 3.2 | 0.2 | 6.7 | 16.7 | 8.6 | 8.6 |
| | NBCE (Su et al., 2024) | 36.2 | 83.1 | 27.9 | 43.3 | 3.8 | 1.3 | 11.7 | 31.0 | 15.9 | 28.2 |
| | CLeHe (Qiu et al.) | 38.4 | 82.6 | 28.4 | 43.6 | 4.2 | 3.2 | 13.4 | 30.8 | 15.8 | 28.9 |
| | DePaC (ours) | 72.8 | 87.4 | 41.6 | 44.8 | 16.7 | 7.5 | 17.3 | 40.7 | 16.4 | 38.4 |
| | ICA (DePaC w/o NegTrain) | 69.7 | 85.1 | 35.9 | 44.2 | 14.5 | 6.2 | 16.2 | 40.1 | 16.1 | 36.4 |
| Llama3-8B | Vanilla (Grattafiori et al., 2024) | 24.3 | 42.3 | 22.3 | 34.6 | 12.6 | 1.6 | 7.2 | 9.6 | 6.4 | 17.9 |
| | AVP (Hao et al., 2022) | 2.1 | 0.4 | 0.2 | 36.9 | 2.9 | 0.4 | 6.9 | 17.3 | 8.2 | 8.4 |
| | NBCE (Su et al., 2024) | 32.8 | 84.2 | 24.8 | 40.3 | 6.5 | 2.1 | 9.9 | 15.6 | 13.9 | 25.6 |
| | CLeHe (Qiu et al.) | 37.2 | 84.0 | 26.2 | 41.7 | 13.3 | 2.7 | 11.5 | 19.6 | 14.3 | 27.8 |
| | DePaC (ours) | 69.5 | 86.6 | 40.2 | 43.9 | 17.4 | 8.2 | 17.6 | 41.0 | 14.1 | 37.6 |
| | ICA (DePaC w/o NegTrain) | 64.8 | 85.0 | 33.8 | 43.2 | 15.2 | 6.8 | 16.4 | 40.3 | 14.0 | 35.5 |
| Phi3-3.8B | Vanilla (Abdin et al., 2024) | 29.7 | 43.5 | 21.2 | 35.7 | 12.3 | 1.3 | 13.2 | 30.2 | 11.3 | 22.0 |
| | AVP (Hao et al., 2022) | 3.4 | 0.3 | 0.5 | 37.9 | 2.3 | 0.7 | 6.3 | 15.9 | 9.4 | 8.5 |
| | NBCE (Su et al., 2024) | 45.4 | 80.3 | 28.3 | 42.2 | 8.6 | 2.2 | 13.8 | 32.5 | 14.7 | 29.8 |
| | CLeHe (Qiu et al.) | 42.2 | 81.2 | 27.6 | 43.6 | 10.1 | 3.8 | 13.1 | 33.1 | 15.7 | 30.0 |
| | DePaC (ours) | 71.4 | 87.0 | 43.2 | 45.3 | 15.5 | 7.2 | 17.5 | 39.1 | 15.3 | 37.9 |
| | ICA (DePaC w/o NegTrain) | 68.6 | 85.2 | 36.3 | 44.5 | 14.0 | 6.1 | 16.5 | 37.9 | 15.1 | 36.0 |

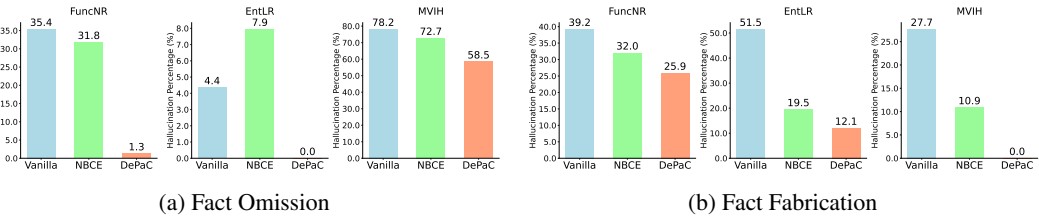

(a) Fact Omission      (b) Fact Fabrication

Figure 5: Hallucination percentage in responses for the information seeking tasks.

The **document-based question-answering (DocQA)** tasks can further reflect how well our DePaC uses the retrieved documents in real-world RAG scenarios. Specifically, we take three real-world long-document tasks to mimic the process of RAG: given a document-specific question, we provide the model several candidate documents, containing one ground-truth document and other unrelated documents. The DocQA tasks include: **Qasper** (Dasigi et al., 2021), **MultifieldQA (MulQA)** (Bai et al., 2023), **NarrativeQA (NarQA)** (Kočiský et al., 2018). Appendix D shows the detailed description of DocQA tasks.

For the evaluation metrics, we use exact-match accuracy in the information seeking tasks and F1 score in the DocQA tasks. On information seeking tasks, we set context window number $k$=8 and evenly divide all items into $k$ windows for all PCE approaches. On DocQA tasks, we augmented the original QA dataset by expanding the number of documents $k$= 5,10,20 in the context. To avoid exceeding window length when concating documents, we treat each document as a context window for PCE approaches.

## 5.2 BASELINES

We compare DePaC with four baselines: **Vanilla**, **AVP** (Hao et al., 2022; Ratner et al., 2023), **NBCE** (Su et al., 2024) and **CLeHe** (Qiu et al.).

- **Vanilla** refers to directly using the vanilla inference approach for a context-limited model (Bai et al., 2023), i.e., concatenating all candidate contexts into input sequence and applying the middle truncation strategy to meet the maximum context length of the model.

- **AVP** (Hao et al., 2022; Ratner et al., 2023) takes the average aggregation (defined in Equation 5) to aggregate the parallel context windows.

- **NBCE** (Su et al., 2024) employs the lowest-uncertainty aggregation (defined in Equation 6) to aggregate the parallel context windows.

Table 2: DocQA results with different candidate document numbers.

| Method | Qasper | | | MulQA | | | NarQA | | |
|---|---|---|---|---|---|---|---|---|---|
| | $k$=5 | $k$=10 | $k$=20 | $k$=5 | $k$=10 | $k$=20 | $k$=5 | $k$=10 | $k$=20 |
| Vanilla (Jiang et al., 2023a) | 15.0 | 13.3 | 8.6 | 39.7 | 33.4 | 31.6 | 10.2 | 9.1 | 9.6 |
| AVP (Hao et al., 2022) | 6.7 | 6.6 | 6.7 | 16.7 | 15.3 | 15.4 | 8.6 | 8.5 | 8.3 |
| NBCE (Su et al., 2024) | 11.7 | 9.9 | 9.8 | 31.0 | 29.0 | 26.9 | 15.9 | 15.8 | 15.1 |
| CLeHe (Qiu et al.) | 13.4 | 10.3 | 10.1 | 30.8 | 28.8 | 26.2 | 15.8 | 15.5 | 14.9 |
| DePaC (ours) | **17.3** | **16.0** | **14.8** | **40.7** | **40.6** | **40.9** | **16.4** | **16.3** | **16.0** |

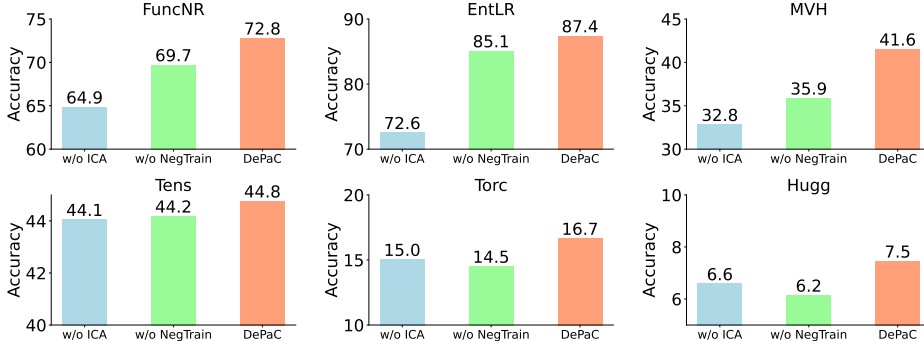

Figure 6: Performance of DePaC without NegTrain or ICA. w/o NegTrain refers to DePaC with positive training, while w/o ICA refers to replace ICA with lowest-uncertainty aggregation of NBCE.

- **CLeHe** (Qiu et al.) ensemble the logits of multiple windows to aggregate the parallel context windows.

## 5.3 MODELS

We conduct experiments on three open-source language models: Mistral-7B (Jiang et al., 2023a), Llama3-8B (Grattafiori et al., 2024) and Phi3-3.8B (Abdin et al., 2024). And we use Mistral-7B (Jiang et al., 2023a) as the default backbone model for the ablation study and analysis.

## 5.4 RESULTS AND ANALYSIS

**DePaC consistently achieves promising performances across nine tasks.** As shown in Table 1, DePaC achieves better performance than baselines across six information seeking tasks and three DocQA tasks. Since the baselines do not require additional training, we also compare solely ICA (DePaC w/o NegTrain) with them in Table 1. The results indicate that using ICA alone outperforms the baselines, and combining ICA with NegTrain further improves performance. The results also show that AVP performs much worse than vanilla. This is because AVP averages the logits across parallel windows, giving equal weight to each window's contribution to the final answer. This makes it underform for RAG scenarios, where it is crucial for the model to identify and focus on the most relevant information from the context.

**DePaC significantly alleviates fact fabrication and fact omission hallucinations.** We analyze the proportion of hallucinations produced by different approaches on three information seeking tasks (FuncNR, EntLR and MVIH). As shown in Figure 5, DePaC significantly reduces the occurrence of both types of hallucinations. DePaC even completely avoids fact omission on EntLR and fact fabrication on MVIH. The detailed hallucination evaluation setup is shown in Appendix G.

**DePaC maintains promising performance with candidate documents number increases.** On DocQA tasks, as the number of documents increases, more redundant information in the context. As

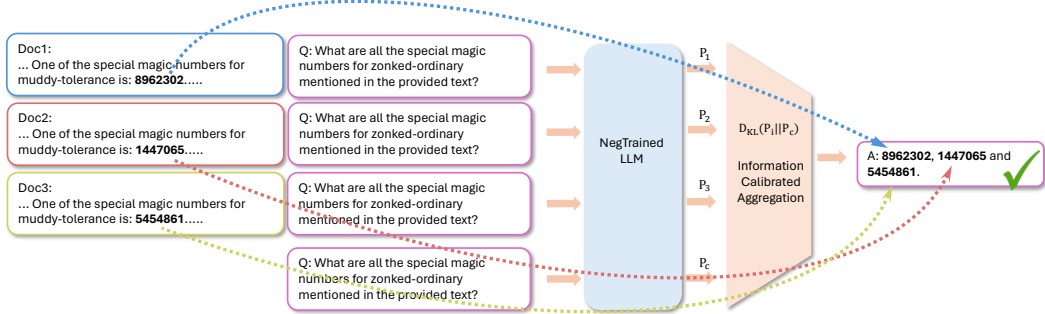

Figure 7: DePaC can switch context window for multi-hop questions.

shown in Table 2, DePaC still achieves promising performance. DePaC's performance with $k$=20 even surpasses NBCE with $k$=5 (23.9 vs. 19.5), further demonstrating DePaC's capability to identify key information from redundant context.

**Both information-calibrated aggregation and context-aware negative training are essential for DePaC performance.** We compare DePaC with two ablation setting: (1) **DePaC w/o NegTrain.** (2) **DePaC w/o ICA**, where we only replace the information-calibrated aggregation function of DePaC with lowest-uncertainty aggregation. We conducte ablation study on the six information seeking datasets. As shown in Figure 6, the ablation results indicate that both parts of DePaC are essential for its performance.

**ICA reduces fact omission, while NegTrain mitigates fact fabrication.** We conduct ablation studies on FuncNR to analyze the effectiveness of NegTrain and ICA in mitigating hallucinations. As shown in the table below, ICA effectively reduces fact omission, while NegTrain mitigates fact fabrication. Combining both ICA and NegTrain yields the best overall performance.

Table 3: Effectiveness of NegTrain and ICA in mitigating hallucinations.

| Method | Fact Omission ↓ | Fact Fabrication ↓ |
|---|---|---|
| Vanilla | 35.4 | 39.2 |
| ICA | 2.7 | 36.8 |
| NegTrain | 33.5 | 27.3 |
| DePaC (ours) | **1.3** | **25.9** |

**DePaC with CoT maintains performance advantage on multi-hop DocQA.** We evaluate on 2WikimQA (Ho et al., 2020) and HotPotQA (Yang et al., 2018) datasets using Mistral-7B. The results in Table 4 show that DePaC still maintains its performance advantage on multi-hop QA datasets. We make the prompt for multi-hop QA datasets end with *"Let's think step by step, "*, this Chain-of-Thought (CoT) prompt (Wei et al., 2022) helps DePaC first

Table 4: Comparison results on multi-hop DocQA tasks.

| Method | 2WikimQA | HotPotQA |
|---|---|---|
| Vanilla (Jiang et al., 2023a) | 19.04 | 12.01 |
| NBCE (Su et al., 2024) | 17.45 | 10.52 |
| CLeHe Qiu et al. | 18.32 | 14.64 |
| DePaC (ours) | **29.72** | **30.95** |

seeks useful information across different contexts before generate the final answer. Figure 7 shows a multi-hop example, where DePaC perform context window switching and successfully locate relevant information spread across multiple documents.

**DePaC also outperforms baselinse on summarization tasks.** We also compare DePac on Mistral-7B with baselines on summarization tasks (GovReport (Huang et al., 2021), QMSum (Zhong et al., 2021), and MultiNews (Fabbri et al., 2019)), which better assess the ability of LLMs to integrate information across entire documents. The results in Table 5 demonstrate that DePaC consistently outperforms the baselines on these summarization tasks.

**DePaC performs better than aggregation approaches for RAG.** We also compare DePaC with previous aggregation approaches specific to RAG (Asai et al., 2023) or can be applied to RAG

Table 5: Comparison results on summarization tasks.

| Method | GovReport | QMSum | MultiNews |
|---|---|---|---|
| Vanilla (Jiang et al., 2023a) | 12.4 | 14.8 | 17.5 |
| NBCE (Su et al., 2024) | 22.3 | 19.6 | 21.3 |
| CLeHe (Qiu et al.) | 22.2 | 20.4 | 21.7 |
| DePaC (ours) | **29.1** | **25.7** | **28.4** |

Table 6: Comparison results between DePaC and aggregation approaches for RAG.

| Method | NaturalQuestions | TriviaQA | RGB |
|---|---|---|---|
| SelfRAG (Asai et al., 2023) | 28.67 | 74.33 | 75.33 |
| CoVe (Dhuliawala et al., 2023) | 26.67 | 68.67 | 76.33 |
| COMPETE (Feng et al., 2024) | 22.67 | 69.00 | 74.00 |
| DePaC (ours) | **33.67** | **88.33** | **94.33** |

(Dhuliawala et al., 2023; Feng et al., 2024), the results in Table 6 show that DePaC outperforms other aggregation approaches on different datasets (Kwiatkowski et al., 2019; Joshi et al., 2017; Chen et al., 2024).

## 6 RELATED WORK

**Retrieval-Augmented Generation (RAG) for LLM.** To address hallucination issue of LLM, Retrieval-augmented generation (Lewis et al., 2020; Gao et al., 2023; Cheng et al., 2024; Asai et al., 2023) has been applied in many fields, including question answering (Zhang et al., 2024), code generation (Zhou et al., 2022; Ma et al., 2024) and recommendation (Zeng et al., 2024). The performance of RAG is limited by the effectiveness of retriever and the information utilization capability of LLM. Some work focus on enhancing the retriever's capabilities (Wang et al., 2023; Lewis et al., 2020). Shi et al. (2024) compresses the retrieved information for LLM. Some work proposes iterative RAG (Jiang et al., 2023b; Shao et al., 2023; Cheng et al., 2024) to help the model progressively utilize document information. Some work (Asai et al., 2023; Dhuliawala et al., 2023; Feng et al., 2024) utilizes prompt engineer to aggregate information from multiple documents to generate a final answer. These methods often lead to information omission during the aggregation process. In this work, we utilize PCE to directly aggregate information from multiple documents when predicting the next token, enhance the accuracy and efficiency of information utilization.

**LLM with Parallel Context Extension (PCE).** Recent research has proposed some PCE approaches to aggregate multiple context windows into a unified representation space, extending context length of LLM. Some research (Hao et al., 2022; Ratner et al., 2023; Li et al., 2024) aggregates by average aggregation mechanisms. Su et al. (2024) proposes NBCE to aggregates by lowest-uncertainty aggregation mechanisms. Previous PCE work primarily focuses on increasing in-context learning examples, and faces hallucination issues when applied for RAG (Yang et al., 2023). Beyond parallel context extension for existing LLM, Yen et al. (2024) also proposes encoder-decoder architecture to implement parallel context. In this work, we propose DePaC to alleviate the hallucination issues of PCE for RAG scenarios. To the best of our knowledge, we are the first work to apply PCE to RAG scenarios.

## 7 CONCLUSION

In this paper, we propose DePaC to address two types of in-context hallucination issues of parallel context extension on RAG. DePaC consists of two key components: (1) a context-aware negative training technique to mitigate fact fabrication, and (2) an information-calibrated aggregation method to address fact omission issue. Both experiments on information seeking and DocQA tasks show the effectiveness of DePaC.

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

This is the Appendix of the paper: *Dehallucinating Parallel Context Extension for Retrieval-Augmented Generation.*

## A    MORE FORMULA DETAILS

The Kullback-Leibler (KL) divergence for discrete probability distributions $\mathbf{P_1}$ and $\mathbf{P_2}$ is defined as:

$$\mathrm{D_{KL}}(\mathbf{P_1} \| \mathbf{P_2}) = \sum_i \mathbf{P_1}(i) \log \frac{\mathbf{P_1}(i)}{\mathbf{P_2}(i)} \tag{13}$$

The cross-entropy loss function is defined as:

$$\mathrm{CE}[p_\theta(\,\cdot\mid d_j \oplus \mathcal{Q}),\,\mathcal{A}] = -\sum_{i=1}^{n} \log p_\theta(\mathcal{A}_i \mid d_j \oplus \mathcal{Q} \oplus \mathcal{A}_{1:i-1}) \tag{14}$$

where $\mathcal{A}_i$ is the $i$-th token in g round-truth answers, $n$ is the sequence length of ground-truth. $p_\theta(\mathcal{A}_i | d_j \oplus \mathcal{Q} \oplus \mathcal{A}_{1:i-1})$ is the probability of generating $\mathcal{A}_i$ given the input $d_j \oplus \mathcal{Q} \oplus \mathcal{A}_{1:i-1}$.

## B    DEPAC SIMPLIFIED FORM

Notice that one implicate constraint in Equation 11 is $\gamma \gg C(\mathbf{p_{i,j}}, \mathbf{p_{i,c}})$ as we hope to directly filter out irrelevant context windows. To simplify this constraint for implementation, we rewrite Equation 11 as the product of two terms and modify Equation 12 to make sure $\hat{C}(\mathbf{p_{i,j}}, \mathbf{p_{i,c}}) \geq 0$,

$$\mathbf{p_i} = \arg\max_{\mathbf{p_{i,j}}} \hat{C}(\mathbf{p_{i,j}}, \mathbf{p_{i,c}}) \cdot \mathbb{I}(\arg\max_k {\mathbf{p_{i,j}}}^k = t_d), \tag{15}$$

$$\hat{C}(\mathbf{p_{i,j}}, \mathbf{p_{i,c}}) = \max_k {\mathbf{p_{i,j}}}^k + \beta \cdot \Delta(\mathbf{p_{i,j}}, \mathbf{p_{i,c}}), \tag{16}$$

where we use $\max_k {\mathbf{p_{i,j}}}^k$ to estimate the output certainty, and $\beta > 0$ is hyper-parameter. For the output of deep learning models, a higher $\max_k {\mathbf{p_{i,j}}}^k$ always indicates a higher certainty in practice (Ghoshal & Tucker, 2022). We set $\beta = 0.2$ by default and analyze the choice of $\beta$ in Appendix E.

## C    INFORMATION SEEKING TASK DETAILS

Below shows the detailed description of information seeking tasks:

- **Function name retrieve (FuncNR)** (An et al., 2024). The contexts in FuncNR contain a large number of Python functions, all of which are sampled from the training data of Starcoder (Li et al., 2023). The questions in FuncNR ask for retrieving the function names based on the given code snippets. We extend the original context length in An et al. (2024) from 32K to 128K.

- **Entity label retrieve (EntLR)** (An et al., 2024). The contexts in EntLR contain a large number of entities, all of which are sampled from Wikidata. Each entity is a triplet in the form of (id, label, description). The questions in EntLR ask for retrieving the labels corresponding to the given entity ids from the contexts. We extend the original context length in An et al. (2024) from 32K to 128K.

- **Multi-values Needle-in-a-Haystack (MVIH)** (Hsieh et al., 2024). The contexts in MVIH contain multiple values for a certain key, along with other unrelated text pieces. The questions in MVIH require the model to seek for all the associated values for the given key.

- **APIBench** (Patil et al., 2023). The contexts in APIBench consist of many real-world APIs, each of which includes an API name, an API call and an API description. The questions in APIBench require to retrieve the API calls based on the given development requirements. Due to the ambiguity in the requirements, APIBench serves as the most challenging evaluation task for information seeking. We take three sub-tasks from APIBench for evaluations: **TensorHub (Tens)**, **TorchHub (Torc)**, and **Huggingface (Hugg)**. In each sub-task, we regard all the candidate APIs as the contexts.

## D    DocQA Task Details

Below shows the detailed description of DocQA tasks:

- **Qasper** (Dasigi et al., 2021). The documents in Qasper are academic research papers and the questions in Qasper are written by NLP practitioners. Specifically, after reading only the title and abstract of each paper, the annotators are required to ask an in-depth question which need the information from the full text to get a comprehensive answer.

- **MultifieldQA** (Bai et al., 2023). The MultifieldQA task aims to test long-document understanding of the model on across diverse fields. The contexts in MultifieldQA are collected from various data sources, including legal documents, government reports, encyclopedias, and academic papers.

- **NarrativeQA** (Kočiskỳ et al., 2018). The NarrativeQA task evaluates how well the model understands the entire long books or movie scripts. Answering the questions in NarrativeQA requires the understanding of the underlying narratives in the given document.

## E    Hyperparameter Settings

We conducted $\beta$ ablation study on the EntLR dataset. The result in Figure 8 indicates that $\beta \in [0.2, 0.3]$ achieves better trade-off between information entropy and KL divergence. We set $\beta = 0.2$ in our experiments.

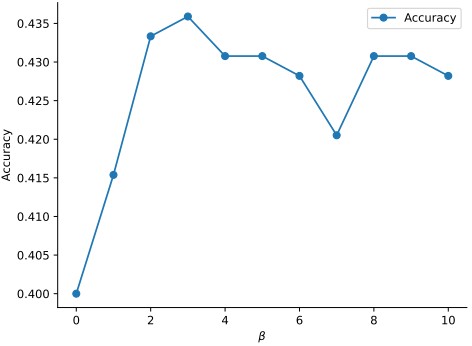

Figure 8: DePaC performance with different $\beta$

## F    Analysis on NegTrain

**Context-aware Negative training can improve the ability of refusing to answer questions with unrelated documents.**    We constructed an additional 4.4K positive samples (PosEval) and negative samples (NegEval), using the same data construction method as NegTrain, but with different seed documents. PosEval represents the situation that documents are related to the question, while NegEval represents the opposite. We compare the rejection token $t_d$ prediction loss on PosEval and NegEval datasets with different NegTrain steps. Figure 9 shows that NegTrain can increase the probability difference between refusing to answer questions with unrelated document and related document.

## G    Hallucination Definition and Evaluation Setup

Previous work (Weng, 2024) categorizes hallucination into two types: (1) **extrinsic hallucination**, where the output of LLM is not grounded by the pre-training dataset or external world knowledge. (2) **in-context hallucination**, where the output of the model is inconsistent with the source content in context. In this work we focus on two types of in-context hallucination: (1) **fact fabrication**, where LLMs present claims that are not supported by the contexts. (2) **fact omission**, where LLMs fail to present claims that are supported by the contexts.

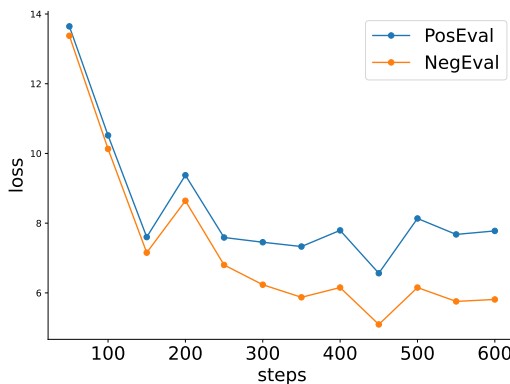

Figure 9: Rejection token prediction loss on PosEval and NegEval over context-aware negative training steps.

| Fact Omission Phrases |
|---|
| not provided, not mentioned, not given, not stated, not available, not included, not specified, not reported, not recorded, not found, not applicable, not clear, not known, not indicated, not listed, not present, not provided, not reported, not shown, not tested, not directly provided, not explicitly mentioned, not explicitly given, cannot be determined, not have a specific, not been mentioned, not contain, not include, not explicitly stated |

Figure 10: Fact omission phrases.

We done in-context hallucination evaluation on three information seeking tasks (FuncNR, EntLR and MVIH), as they are evaluated by exact-match score, makes them easier to analyze than QA tasks. Since these tasks have clear answers in the document and all incorrect outputs are hallucinations, we manually analyzed the data to define 27 fact omission phrases (shown in Figure 10), counted the incorrect outputs that appeared with these phrases as fact omission, and classified other errors as fact fabrication.

## H  WINDOW NUMBER ANALYSIS

To analyze DePaC's performance with different numbers of windows, we conduct experiments on the FuncNR dataset, keeping the total number of candidate functions constant while varying the number of windows into which the context is divided. The results in Figure 11 show that as the number of windows increases (form 4 to 128), DePaC's information-seeking ability improves; however, when the number of windows becomes too large (larger than 256), there may be a slight performance decline. All DePaC with split-window outperforms the single-window, further validating the effectiveness of DePaC with parallel context windows.

## I  EFFECTIVENESS OF NEGTRAIN

As shown in Table 7, to further show the effectiveness of NegTrain, we compare NegTrain-Llama2-13B with SlefRAG-Llama2-13B Asai et al. (2023) (which enhance model's ability of abstaining

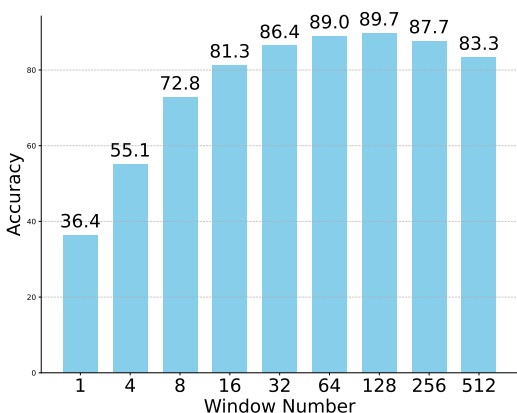

Figure 11: DePaC performance at different degrees of context window parallelism.

Table 7: FactCheckQA results.

| Model | FactCheckQA |
|---|---|
| Llama2-13B-Chat Touvron et al. (2023) | 73 |
| SlefRAG-Llama2-13B Asai et al. (2023) | 76.5 |
| NegTrain-Llama2-13B | **78.5** |

irrelevant information from context) on FactCheckQA Bashlovkina et al. (2023) benchmark (which requires LLM to answer the question based on the provided context). The results show that NegTrain outperforms SelfRAG and original Llama2 model on FactCheckQA dataset.

## J  BROADER IMPACTS

This work used GPT-4-Turbo to generate training data. Therefore, our fine-tuned model may inherit the potential risks of GPT-4-Turbo in terms of ethical and safety issues.

## K  LIMITATIONS

**Data generation cost.** We rely on GPT-4-Turbo to generate our training data, which cost around 90$ for API calling. Future work should attempt to generate data using cheaper models without compromising data quality.

**Training cost.** Our training process consumes some computational resources, but it's a one-time effort. Given the advantages of our method in terms of inference efficiency and accuracy, we believe these offline costs are justified.

## L  FUTURE WORK

As shown in Figure 1, though our DePaC significantly reduces the occurrence of hallucinations in responses, the hallucination phenomenon still exists. For example, in some scenarios, both windows may contain relevant content, but only one is helpful for answering the question. DePaC may mistakenly select the relevant but unhelpful window. LLMs may fail to utilize useful information even within windows containing relevant documents. Combining DePaC with previous work Xiong et al. (2023); An et al. (2024) that enhances LLMs' ability to processing context should further improve DePaC's performance.

