# OpenReview forum: "Dehallucinating Parallel Context Extension for Retrieval-Augmented Generation"
_ICLR.cc/2026/Conference — ICLR 2026 Conference Withdrawn Submission_

### Official Review · Reviewer_9pyA · 2025-10-28

**Soundness:** 3
**Presentation:** 4
**Contribution:** 2
**Rating:** 4
**Confidence:** 4

**Summary:**

This paper introduces a method called DePac, designed to reduce hallucinations when applying Parallel Context Extension (PCE) in retrieval-augmented generation.
The authors identify two major problems in this setting: (1) fact fabrication and (2) fact omission.
To address the first problem (fact fabrication), they propose Context-Aware Negative Training (NegTrain), which trains the model to abstain from answering when it determines that a question is unanswerable.
To tackle the second problem (fact omission), they introduce Informed-Calibrated Aggregation (ICA), which modifies the uncertainty measurement by computing the information difference between using and not using a specific context.
In their complexity analysis, the authors claim that the proposed method is more efficient than existing baselines.
Experimental results on six information-seeking tasks and three document-based question-answering tasks demonstrate that DePac achieves competitive performance compared to other methods for parallel context extension.

**Strengths:**

- The paper is well-structured and clearly presents a new approach (though it appears to be a combination of two existing methods) for parallel context extension.
- Compared with the baselines, the proposed method consistently achieves performance improvements across different models and tasks.
- The authors also include a complexity analysis, demonstrating that the proposed method is more efficient than the naïve approach of directly concatenating documents as input.

**Weaknesses:**

- The proposed method appears to be a relatively straightforward combination of two existing ideas: (1) learning to abstain or refuse, and (2) aggregating information from parallel context extensions (PCE) using a contrastive decoding–like algorithm. Beyond this incremental integration, the paper provides limited discussion of related work—particularly methods concerning learning to refuse and contrastive decoding, such as [1] and [2]. Expanding the related work section to position DePac more clearly within this existing research landscape would improve the paper’s contribution and clarity.
- The complexity analysis may be somewhat overstated. The reported difference between the original and proposed approaches—O(k^2) versus O(k)—is arguably minor, since k typically ranges only in the tens for most RAG scenarios and can thus be treated as a constant rather than a true variable. Consequently, the practical computational gain might be limited. Moreover, the analysis does not account for the overhead introduced by the parallel nature of DePac, such as the cost of aggregating results from parallel computations. These factors make it unclear whether DePac is indeed more efficient in realistic settings.
- A more careful comparison with stronger sequential baselines would strengthen the empirical validity of the proposed approach. Given the continuous improvement of large language models capable of handling longer contexts, the advantages of parallel computation may diminish relative to sequential methods. It would therefore be valuable to compare DePac fairly against competitive sequential baselines under the same computational budget, rather than primarily against relatively simple parallel baselines.

References

[1] R-Tuning: Instructing Large Language Models to Say “I Don’t Know.”

[2] Trusting Your Evidence: Hallucinate Less with Context-Aware Decoding.

**Questions:**

Please see the Weaknesses section.

---

### Official Review · Reviewer_fdJj · 2025-10-30

**Soundness:** 3
**Presentation:** 3
**Contribution:** 2
**Rating:** 4
**Confidence:** 4

**Summary:**

The paper focuses on how to process multiple retrieved documents in parallel (PCE) to avoid the long context issue caused by concatenating different documents in traditional RAG systems. It proposes DePaC, which aims to address the problems of fact fabrication and fact omission. In the first stage, DePaC trains the model to output “unknown” when faced with irrelevant contexts, in order to avoid fact fabrication. In the second stage, it selects which document to utilize based on the change in the model’s token probability distribution before and after incorporating each document. Overall, DePaC achieves improvements compared to traditional PCE methods.

**Strengths:**

1. This paper studies an important question — how to process multiple RAG documents in parallel rather than concatenating them, in order to improve efficiency.

2. The paper proposes DePaC, which includes two approaches. First, it trains the model to say “I don’t know” when faced with irrelevant documents. Second, it introduces ICA, which determines which document to use for decoding the current token based on the magnitude of change each document brings. The idea is quite interesting.

3. The experiments in the paper are thorough (although I think some parts may be somewhat questionable).

4. The writing of the paper is fairly good.

**Weaknesses:**

1. In the first paragraph of the Introduction, I’m not fully convinced by the assumption that the retrieved documents are independent. For multi-hop questions, it often requires reasoning across multiple complementary documents rather than treating them in isolation.

2. The term vanilla method appears without explanation. It would help readers if you could briefly clarify what it refers to when it first appears.

3. The example in Figure 7 may not be the most illustrative one. For instance, a question like “Who is the daughter of the President of the United States?” better demonstrates the need for multi-hop reasoning — first identifying the president, then determining his daughter. It would be useful to clarify whether PCE can handle such cases.

4. The experimental setup might be somewhat idealized, as it provides one ground-truth document and several unrelated ones. This setup may not fully reflect the complexity of real-world RAG scenarios.

5. I have some reservations about the ICA approach. As I understand it, ICA measures how much the model’s output changes before and after adding a document. However, such change does not always correspond to resolving fact omission. For example, if the model initially answers correctly but produces “unknown” after adding an irrelevant document, the change would still be large but the performance worse. This suggests that ICA’s assumption—that greater change indicates better answerability—might not always hold.

**Questions:**

If the model outputs “unknown” after incorporating a document, does that mean this document can be excluded during token logits aggregation?

---

### Official Review · Reviewer_PEJ2 · 2025-10-30

**Soundness:** 3
**Presentation:** 2
**Contribution:** 2
**Rating:** 4
**Confidence:** 3

**Summary:**

This paper focuses on solving the in-context hallucination problem of Parallel Context Extension (PCE) in Retrieval-Augmented Generation (RAG) scenarios. The authors propose DePaC, which includes two core components—Context-aware Negative Training to guide models to refuse answering when contexts are unrelated and Information-Calibrated Aggregation to and prioritize high-information windows. Experiments on varies datasets show that DePaC outperforms baselines by on average accuracy, significantly reduces both types of hallucinations.

**Strengths:**

1. The authors proposed a effective method DePaC to address two hallucination (fact fabrication and fact omission) of Parallel Context Extension (PCE) in Retrieval-Augmented Generation (RAG) scenarios.

2. The authors conduct comprehensive experiments across nine tasks, encompassing information retrieval, long-document question answering, multi-hop question answering, and summarization, which effectively demonstrate the proposed method’s efficacy in mitigating hallucinations in retrieval-augmented generation (RAG) systems.

**Weaknesses:**

1. It is recommended that the authors further conduct theoretical or empirical analyses on the two types of hallucinations—fact fabrication and fact omission. Although illustrative examples for each type are provided, a more detailed analysis distinguishing their underlying causes is lacking. Furthermore, in the context of retrieval-augmented generation (RAG), do all hallucinations fall within these two categories, or might there exist other types of hallucinations? Can the proposed method effectively address such potential additional hallucination types?

2. The authors are encouraged to clarify the novelty of the proposed method relative to existing techniques. Specifically, Context-aware Negative Training appears to primarily involve constructing positive and negative samples for contrastive fine-tuning. Moreover, the distinction of Information-Calibrated Aggregation from prior RAG approaches based on information gain or entropy [1][2] should be explicitly articulated.

3. Given the specific characteristics of PCE tasks, it is advisable that the authors supplement their study with experiments involving a larger number of documents (thereby extending the context length) and additional models or frameworks with enhanced long-context capabilities, in order to further demonstrate the practical applicability of the proposed method in PCE scenarios.

[1] Entropy-Based Decoding for Retrieval-Augmented Large Language Models. NAACL

[2] GainRAG: Preference Alignment in Retrieval-Augmented Generation through Gain Signal Synthesis. ACL 2025

**Questions:**

1.The Context-aware Negative Training proposed in the paper enables the model to refuse answering when contexts are unrelated; however, it remains unclear whether this behavior impacts the model’s general reasoning capabilities. A theoretical or empirical analysis addressing this potential effect is warranted.

2.Regarding NegTrain’s negative document sampling, it is important to clarify how "irrelevance" is defined. Is it determined based on semantic similarity (e.g., cosine similarity below a certain threshold between document and question embeddings), or is it obtained via random sampling?

3.Additionally, under certain dataset settings, the proposed method does not achieve state-of-the-art performance compared to existing approaches. It is recommended that the authors provide further analysis to elucidate the underlying reasons for this performance gap.

4.How do the authors ensure the accuracy of the data generated using GPT-4-Turbo? Has any human verification or validation been conducted?

If the authors can adequately address all of the concerns raised, I am willing to consider raising my final rating.

---

### Official Review · Reviewer_5juz · 2025-10-31

**Soundness:** 3
**Presentation:** 3
**Contribution:** 2
**Rating:** 6
**Confidence:** 3

**Summary:**

This paper introduces Dehallucinating Parallel Context Extension (DePaC), a dehallucination-oriented framework for retrieval-augmented generation (RAG). By integrating context-aware negative training and information-calibrated aggregation, DePaC effectively mitigates two major types of in-context hallucinations—fact fabrication and fact omission—that commonly occur in parallel context extension (PCE) settings. Without modifying the underlying model architecture, DePaC enables large language models to more reliably utilize retrieved knowledge while maintaining linear inference efficiency. Extensive experiments across nine RAG benchmarks demonstrate that DePaC significantly reduces hallucinations and achieves state-of-the-art performance in both single- and multi-hop reasoning tasks.

**Strengths:**

1. In terms of originality, the work is one of the first to explicitly address hallucination issues within the Parallel Context Extension (PCE) paradigm for Retrieval-Augmented Generation (RAG).

2. Regarding technical quality, the methodology is well-motivated and technically sound.     The paper provides clear mathematical formulations, including a well-defined KL-based information increment measure and an interpretable integration into the aggregation objective.

3. The experimental design is comprehensive: it covers nine RAG benchmarks, includes three different backbone models, and performs ablation, efficiency, and hallucination-type analyses.

**Weaknesses:**

1.  The innovation of this paper lies in its reasonable thinking, but it relies partly on the combination of existing concepts rather than introducing completely new principles.  Its two key components: negative sample training with rejection labels and information-based aggregation - were all inspired by previous research.  The concept of "rejection response" has been discussed in abandonment learning and illusion mitigation, while the KL-based information gain index appears in uncertainty calibration.

2.  The construction of the negative sample training dataset raises concerns about ecological validity.  Training with question-answer pairs generated by GPT-4-Turbo from C4 and randomly selected negative samples may not fully reflect the complex noise patterns in the real retrieval scenarios.

3.  This work still has room for improvement in the interpretability and transparency of hyperparameters.  The coefficients β and γ in the aggregation target have a significant impact on the sensitivity selection, but their adjustment process is only briefly mentioned in the appendix.

**Questions:**

1. Could the author clarify more clearly the differences between this study and the existing negative sample SFT methods [1] as well as the selection methods based on information gain [2][3]?

2. The construction of negative sample data relies on GPT-4-Turbo to generate question-answer pairs, and randomly irrelevant texts are used as negative samples.  What are the results when the negative sample part is relevant with the query?

3. The simplified implementation theory formulas (Formulas 11 - 12) can be found in Appendix B. The author is recommended  to compare the performance differences between the complete formula and the simplified version and explain whether this simplification is due to stability considerations or numerical issues.

4. The measurement of hallucinations in this paper is one of its core arguments, but the description of the measurement method in the main text is rather brief.

5. How does ICA perform when there are contradictory evidence among documents?

[1] Direct Preference Optimization

[2] Pointwise Mutual Information as a Performance Gauge for RAG

[3] RAGRAPH: A General Retrieval-Augmented Graph Learning Framework

---

### Note · Authors · 2025-11-26

I have read and agree with the venue's withdrawal policy on behalf of myself and my co-authors.